# Articulation points in complex networks

Liang Tian[1,2], Amir Bashan[3], Da-Ning Shi[2] & Yang-Yu Liu[1,4]

An articulation point in a network is a node whose removal disconnects the network. Those nodes play key roles in ensuring connectivity of many real-world networks, from infrastructure networks to protein interaction networks and terrorist communication networks. Despite their fundamental importance, a general framework of studying articulation points in complex networks is lacking. Here we develop analytical tools to study key issues pertinent to articulation points, such as the expected number of them and the network vulnerability against their removal, in an arbitrary complex network. We find that a greedy articulation point removal process provides us a different perspective on the organizational principles of complex networks. Moreover, this process results in a rich phase diagram with two fundamentally different types of percolation transitions. Our results shed light on the design of more resilient infrastructure networks and the effective destruction of terrorist communication networks.

[1] Channing Division of Network Medicine, Brigham and Women's Hospital and Harvard Medical School, Boston, Massachusetts 02115, USA. [2] College of Science, Nanjing University of Aeronautics and Astronautics, Nanjing 210016, China. [3] Physics Department, Bar-Ilan University, Ramat-Gan 5290002, Israel. [4] Center for Cancer Systems Biology, Dana Farber Cancer Institute, Boston, Massachusetts 02115, USA. Correspondence and requests for materials should be addressed to Y.-Y.L. (email: yyl@channing.harvard.edu).

A fundamental challenge in studying complex networked systems is to reveal the interplay between network structure and function[1,2]. Here we tackle this challenge by investigating a classical notion in graph theory, that is, articulation points. A node in a network is an articulation point (AP) if its removal disconnects the network or increases the number of connected components in the network[3,4] (Fig. 1a). Those APs can be easily identified using a linear-time algorithm based on depth-first search[5]. It has been found that APs play important roles in ensuring the robustness and connectivity of many real-world networks. For example, in infrastructure networks such as air traffic networks or power grids, APs, if disrupted or attacked, pose serious risks to the infrastructure[6,7]. In wireless sensor networks, failures of APs will block data transmission from one network component to others[8]. In the yeast protein–protein interaction network, lethal mutations are enriched in the group of highly connected proteins that are APs[9]. Analysis of APs hence provides us a different angle to systematically investigate the structure and function of real-world networks.

Despite the importance of APs in ensuring the robustness and connectivity of many real-world networks, we still lack a deep understanding on the roles of APs in many complex networks. Can we design an AP-based attack strategy to more efficiently destroy malicious networks? Can we develop an AP-based network decomposition method to better reveal the organizing principles of complex networks? What happens if we keep removing APs from a random graph or a real network? Will there be a core left? If yes, what's the implication of such a core in terms of structural integrity and functionality of the network? How to quantify if a real network has overrepresented or under-represented APs comparing to its randomized counterparts? In this article we offer an analytical framework to study those fundamental issues pertinent to APs in both real networks and random graphs, harvesting a series of interesting results.

## Results

**Articulation point–targeted attack**. Representing natural vulnerabilities of a network, APs are potential targets of attack if one aims for immediate damage to a network. Note that the removal of an AP in a network may lead to the emergence of new APs in the remainder of the network, that is, new potential targets of attack (Fig. 1b). This fact inspires us to design a brute-force AP-targeted attack (APTA) strategy: iteratively remove the most destructive AP that will cause the most nodes disconnected from the giant connected component (GCC) of the current network. Given a limited 'budget' (that is, the number of nodes to be removed), this APTA strategy is very efficient in reducing the GCC, compared with strategies based on other node centrality measures, such as degree[10,11] and collective influence[12]. Indeed, we find that for a small fraction of removed nodes APTA leads to the fastest reduction of GCC for a wide range of real-world networks from technological to infrastructure, biological, communication, and social networks (Supplementary Note 1; Supplementary Fig. 1). Depending on the initial network structure, APTA would either completely decompose the network or result in a residual GCC that occupies a finite fraction of the network. This residual GCC is a biconnected component (or bicomponent), in which any two nodes are connected by at least two independent paths and hence no AP exists[5]. For accuracy, we will call it residual giant bicomponent (RGB) hereafter. This RGB naturally represents a core that maintains the structural integrity of the network.

**Greedy articulation points removal**. We also find that the identification and removal of APs provide us a new perspective on the organizational principles of complex networks. For example, in the terrorist communication network of the 9/11 attacks on U.S. (Fig. 1a), each AP member (shown in red) can be considered as a messenger of a particular subnetwork, because any information exchange between that subnetwork and the rest of the network passes through the AP[13]. All the APs and their associated subnetworks in the original network constitute the first layer of the terrorist network. After removing all the APs in the original network, the first layer is peeled off, new APs emerge and the second layer of the network is exposed. We can repeat this greedy APs removal (GAPR) process until there is no AP left in the network. Note that at each step we simultaneously remove all the APs present in the current network. Figure 1 illustrates this network decomposition process in the 9/11 terrorist communication network, which has 62 terrorists. We find that this network consists of three layers and an RGB of 26 nodes. (Note that the RGB associated with the GAPR process is similar to but not necessarily the same as that of the APTA process, see Supplementary Note 2; Supplementary Fig. 2.) Interestingly,

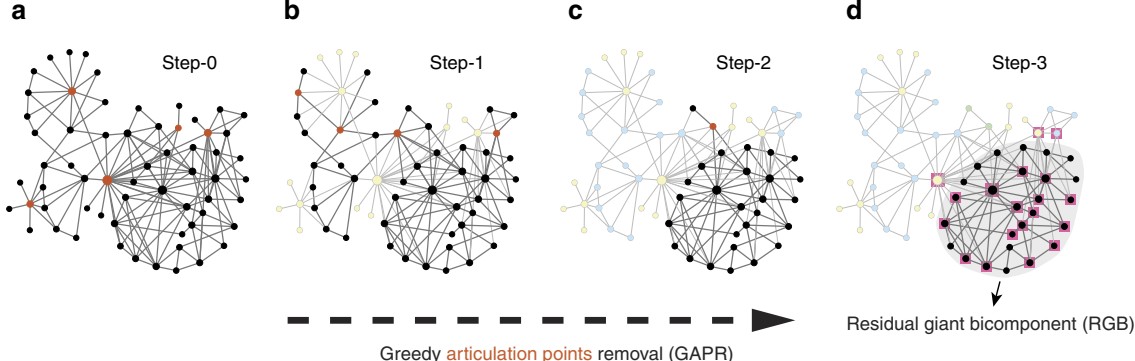

**Figure 1 | Articulation points and the greedy articulation points removal process.** (**a**) Articulation points in the terrorist communication network from the attacks on the United States on September 11, 2001 are highlighted in red. This network contains in total 62 nodes and 153 links[13]. (**b,c**) At each time step, all the articulation points and the links attached to them are removed from the network. This greedy articulation points removal procedure can be considered as a network decomposition process: at each step, all the removed nodes (because of the removal of articulation points in the current network) form a layer in the network. We peel the network off one layer after another, until there is no articulation point left. We find that this terrorist communication network consists of 3 layers, shown in light yellow, blue, and green, respectively. (**d**) After 3 steps, a well-defined residual giant bicomponent is left, which contains 26 of the 62 nodes. Interestingly, 16 of the 19 hijackers (highlighted with squares) are in the residual giant bicomponent, which is statistically significant (Fisher's exact test yields a two-tailed test $P$ value $1.13 \times 10^{-5}$).

among those 26 RGB nodes, 16 of them are hijackers in the 9/11 terrorist attack, which in total has 19 hijackers. In a sense, this RGB serves as a core maintaining the functionality of this covert network, which has a particular goal—hijacking. Note that some of the hijackers in the RGB are not hubs (that is, highly connected nodes), but only have two or three neighbours in the network. Hence they cannot be easily identified through traditional network decomposition methods, for example, maximum clique[3,4], $k$-core decomposition[14,15] and $t$-core decomposition[16–18], which are designed to uncover or extract a dense core structure consisting of highly connected nodes (Supplementary Note 3; Supplementary Fig. 3).

Interestingly, we find that the two RGBs associated with the APTA and GAPR processes significantly overlap for many real-world networks (Supplementary Note 2; Supplementary Fig. 2). Note that, compared with APTA, the GAPR process is deterministic and avoids the optimization of the damage caused by nodes removal, which make it analytically solvable. Hereafter we focus on the RGB obtained from the GAPR process.

**Articulation points and residual giant bicomponent in real networks.** The results presented in the previous subsections prompt us to study the fraction of APs ($n_{AP} := N_{AP}/N$) and the relative size of the RGB ($n_{RGB} := N_{RGB}/N$) in a wide range of real-world networks. Here $N_{AP}$, $N_{RGB}$ and $N$ represent the number of APs, the number of nodes in the RGB (obtained from the GAPR process), and the number of nodes in the whole network, respectively.

We find that many real networks have a non-ignorable fraction of APs and a rather small RGB (Fig. 2a). One may expect that infrastructure networks should have a relatively small fraction of APs and a large RGB, and hence are very robust against AP removal. Interestingly, this is not the case. The power grids in two regions of U.S. have almost the largest fraction ($\sim 24\%$) of APs among all the real networks analysed in this work. And they have almost no RGB. The road networks of three states in U.S. have almost 20% of APs, and a small RGB ($n_{RGB} \sim 0.08$). These results suggest that infrastructure networks are apparently not optimized with respect to AP removal. Indeed, because of the high cost of adding new links (for example, connecting two power stations with high-voltage transition lines, or connecting two cities with a new highway), infrastructure networks typically lack a high redundancy, but are often optimized with respect to other criteria, such as social profitability. By contrast, among all the 28 food webs we analysed, 22 of them have no APs (and hence $n_{RGB} = 1$). In other words, those ecological networks tend to be biconnected and the extinction of one species will not disconnect the whole community. This high structural robustness could be because of evolutionary inter-species interactions across the whole community[19].

More interestingly, we find that most of the real networks analysed here have either a very small RGB or a rather big one (see Fig. 2a, light magenta and turquoise regions). Later we will show that this phenomenon is related to a discontinuous phase transition associated with the GAPR process.

To identify the topological characteristics that determine these two quantities ($n_{AP}$ and $n_{RGB}$), we compare $n_{AP}$ (or $n_{RGB}$) of a given real network with that of its randomized counterpart. To this aim, we randomize each real network using a complete randomization procedure that turns the network into an Erdős-Rényi (ER) type of random network with the number of nodes

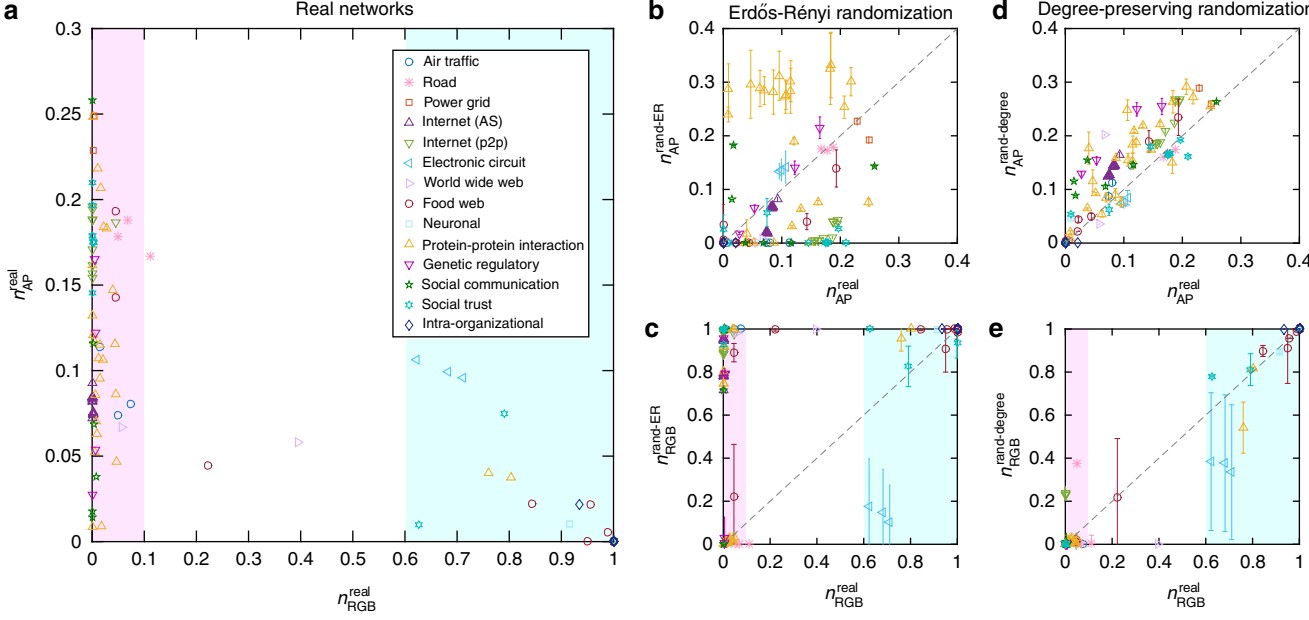

**Figure 2 | Articulation points and the residual giant bicomponent in real networks.** (**a**) Fraction of articulation points ($n_{AP}^{real}$) versus relative size of the residual giant bicomponent ($n_{RGB}^{real}$) is plotted for a wide range of real networks, from infrastructure networks to technological, biological, and social networks. Most of the real networks analysed here have either a very small residual giant bicomponent or a rather big one (highlighted in light magenta and turquoise, separately). (**b,c**) Fraction of articulation points ($n_{AP}^{rand-ER}$) and relative size of the residual giant bicomponent ($n_{RGB}^{rand-ER}$), obtained from the fully randomized counterparts of the real networks, compared with the exact values ($n_{AP}^{real}$ and $n_{RGB}^{real}$). (**d,e**) Fraction of articulation points ($n_{AP}^{rand-degree}$) and relative size of the residual giant bicomponent ($n_{RGB}^{rand-degree}$), calculated from the degree-preserving randomized counterparts of the real networks, compared with the exact values ($n_{AP}^{real}$ and $n_{RGB}^{real}$). In **b–e**, all data points and error bars (standard error of the mean or s.e.m.) are determined from 100 realizations of the randomized networks, and the dashed lines ($y = x$) are guide for eyes. For detailed description of these real networks and their references, see Supplementary Note 7; Supplementary Tables 1–14.

$N$ and links $L$ unchanged[20]. We find that most of the completely randomized networks possess very different $n_{AP}$ (or $n_{RGB}$), comparing to their corresponding real networks (Fig. 2b,c). This indicates that complete randomization eliminates the topological characteristics that determine $n_{AP}$ and $n_{RGB}$. By contrast, when we apply a degree-preserving randomization, which rewires the links among nodes, while keeping the degree $k$ of each node unchanged, this procedure does not alter $n_{AP}$ and $n_{RGB}$ significantly (Fig. 2d,e). In other words, the characteristics of a network in terms of $n_{AP}$ and $n_{RGB}$ is largely encoded in its degree distribution $P(k)$. Most of the real-world networks display slightly smaller $n_{AP}$ and bigger $n_{RGB}$ than their degree-preserving randomized counterparts. We attribute these differences to higher-order structure correlations, such as clustering[21] and degree assortativity[22], which are eliminated in the degree-preserving randomization.

**Analytical framework of the greedy articulation points removal process.** The results of $n_{AP}$ and $n_{RGB}$ in real-world networks encourage us to analytically calculate $n_{AP}$ and $n_{RGB}$ for networks with prescribed degree distributions[23]. To achieve that, we analyse the GAPR process on infinitely large networks and explore in depth the effect of different degree distributions on $n_{AP}$ and $n_{RGB}$. Consider the discrete-time dynamics of the deterministic GAPR process, which generates a series of snapshots for the remainder network with a clear temporal order $\{0, 1,..., t,...., T\}$. Here, $T$ is the total number of GAPR steps, which is also the number of layers peeled off during the GAPR process. We denote the fraction of APs and the relative size of the GCC in the original network as $n_{AP}(0)$ and $n_{GCC}(0)$, respectively. Removal of the original APs leads to a new fraction of APs $n_{AP}(1)$ and a smaller GCC of relative size $n_{GCC}(1)$. We repeat this process and denote the fraction of APs and the relative size of the GCC of the network snapshot at time step $t$ as $n_{AP}(t)$ and $n_{GCC}(t)$, respectively. At the end of the GAPR process, we have $n_{AP}(T) = 0$ and $n_{GCC}(T) = n_{RGB}$. On the basis of the configuration model of uncorrelated random networks[23–25], we can analytically calculate $n_{AP}(t)$ and $n_{GCC}(t)$ for networks with arbitrary degree distributions at any time step $t$. This enables us to further compute $T$ and $n_{RGB}$. See Methods section and Supplementary Note 4 for the details of our analytical framework of the GAPR process.

**Articulation points in classical model networks.** The analytical framework of the GAPR process enables us to calculate various quantities of interests. We first investigate the fraction of APs in the original network, that is, $n_{AP} = n_{AP}(0)$. We calculate $n_{AP}$ in two canonical model networks: (1) ER random networks with Poisson degree distributions $P(k) = e^{-c}c^k/k!$, where $c$ is the mean degree (hereafter we also use $c$ to denote the mean degree of a general network); and (2) scale-free (SF) networks with power-law degree distributions $P(k) \sim k^{-\lambda}$, where $\lambda$ is often called the degree exponent (Fig. 3a,b). The fraction of APs is trivially zero in the two limits $c \to 0$ and $c \to \infty$, and reaches its maximum at a particular mean degree $c_{AP}$. For ER networks, we find that $c_{AP} = 1.41868\cdots$, which is larger than $c_p = 1$, the critical point of ordinary percolation where the GCC emerges[1,2].

The phenomenon that $n_{AP}$ displays a unimodal behaviour and the fact that $c_{AP} > c_p$ can be explained as follows. The process of increasing the mean degree $c$ of an ER network can be considered as the process of randomly adding links into the network. When the mean degree $c$ is very small (nearly zero), there are only isolated nodes and dimers (that is, components consisting of two nodes connected by one link), and thus $n_{AP} \to 0$. With $c$ gradually increasing but still smaller than $c_p$, the network is full of finite connected components (FCCs), most of which are trees (Fig. 3c).

Hence, in the range of $0 < c < c_p$ most of the nodes (except isolated nodes and leaf nodes) are APs, and adding more links to the network will increase the number of APs (Fig. 3c). When $c > c_p$, the GCC develops and occupies a finite fraction of nodes in the network (highlighted in light blue in Fig. 3d–f). In this case, we can classify the links to be added to the network into two types: (I) links inside the GCC (yellow dashed lines); and (II) links that connect the GCC with an FCC or connect two FCCs (turquoise dashed lines). The probability that an added link is type-I (or type-II) as a function of the mean degree $c$ is shown in Fig. 3a (light blue region). Adding type-I links to the network will never induce new APs, and may even convert the existing APs (see Fig. 3d,e, nodes in black boxes) back to normal nodes. By contrast, adding type-II links will never decrease the number of APs and could convert normal nodes (see Fig. 3d,e, nodes in orange boxes) to APs. The contributions of these two types of links to $n_{AP}$ compete with each other. At the initial stage of this range of $c$ ($c > c_p$), since the GCC is still small, most of the added links are type-II (Fig. 3a, turquoise dashed line), and thus $n_{AP}$ continues to increase (Fig. 3a, red line). At certain point $c_{AP}$, where the peak of $n_{AP}$ locates, the contribution of type-I links to $n_{AP}$ overwhelms that of type-II links, hence $n_{AP}$ begins to decrease. When the mean degree $c$ is large enough, the network itself becomes a bicomponent without any AP.

The phenomena that $c_{AP} > c_p$ is even more prominent for SF networks generated by the static model[26], where $c_p(\lambda) < 1$ and $c_{AP}(\lambda) > 1.41868\cdots$. This is because, for those SF networks, even though the GCC emerges at lower $c_p(\lambda)$, its relative size is rather small at the initial stage of its emergence and the network is more fragmented in FCCs, which results in larger $c_{AP}(\lambda)$ (Fig. 3b) (see Supplementary Note 5 for details).

**Percolation transitions associated with greedy articulation points removal.** We now systematically study the behaviours of $n_{GCC}(t)$ and $n_{RGB}$, as functions of the mean degree $c$ for infinitely large ER networks. To emphasize the $c$-dependence, hereafter we denote $n_{GCC}(t)$ and $n_{RGB}$ of ER networks as $n_{GCC}(t, c)$ and $n_{RGB}(c)$, respectively. To systematically characterize the percolation transitions, various quantities will be analysed, such as the critical mean degree, critical exponents, the jump size of the order parameter at criticality, and so on.

As shown in Fig. 4a (grey lines), after any finite steps of GAPR, the GCC always emerges in a continuous manner, suggesting a continuous phase transition. Hereafter we will call it GCC percolation transition. For $t$ steps of GAPR, the GCC percolation transition displays a critical phenomenon: $n_{GCC}(t, c) \sim (c - c^*(t))^{\beta(t)}$ for $(c - c^*(t)) \to 0^+$, where $c^*(t)$ is the critical mean degree, that is, the percolation threshold, and $\beta(t)$ is the critical exponent. We find that as $t$ increases, $c^*(t)$ becomes larger and larger, but eventually converges to $c^*(\infty) = c^* = 3.39807\cdots$. Note that, for any finite $t$, the critical exponent $\beta(t)$ associated with the GCC percolation transition is the same: $\beta(t) = \beta_{GCC} = 1$ (see Fig. 4c, grey lines).

By contrast, if we allow for infinite steps of GAPR (that is, we stop the process only if there is no AP left), the size of the resulting GCC (that is, the RGB), denoted as $n_{RGB}(c)$, displays a remarkable discontinuous phase transition: $n_{RGB}(c)$ abruptly jumps from zero (when $c < c^*$) to a finite value at $c^*$, and then increases with increasing $c$ (see Fig. 4a black line). Hereafter we will call it RGB percolation transition. If we denote the jump size as $\Delta$, we find that $n_{RGB}(c) - \Delta \sim (c - c^*)^{\beta_{RGB}}$ with critical exponent $\beta_{RGB} = 1/2$ when $(c - c^*) \to 0^+$ (Fig. 4c, black line), suggesting that the RGB percolation transition is actually a hybrid phase transition[14,27,28]. In other words, $n_{RGB}(c)$ has a jump at the critical point $c^*$ as a first-order phase transition but also has a

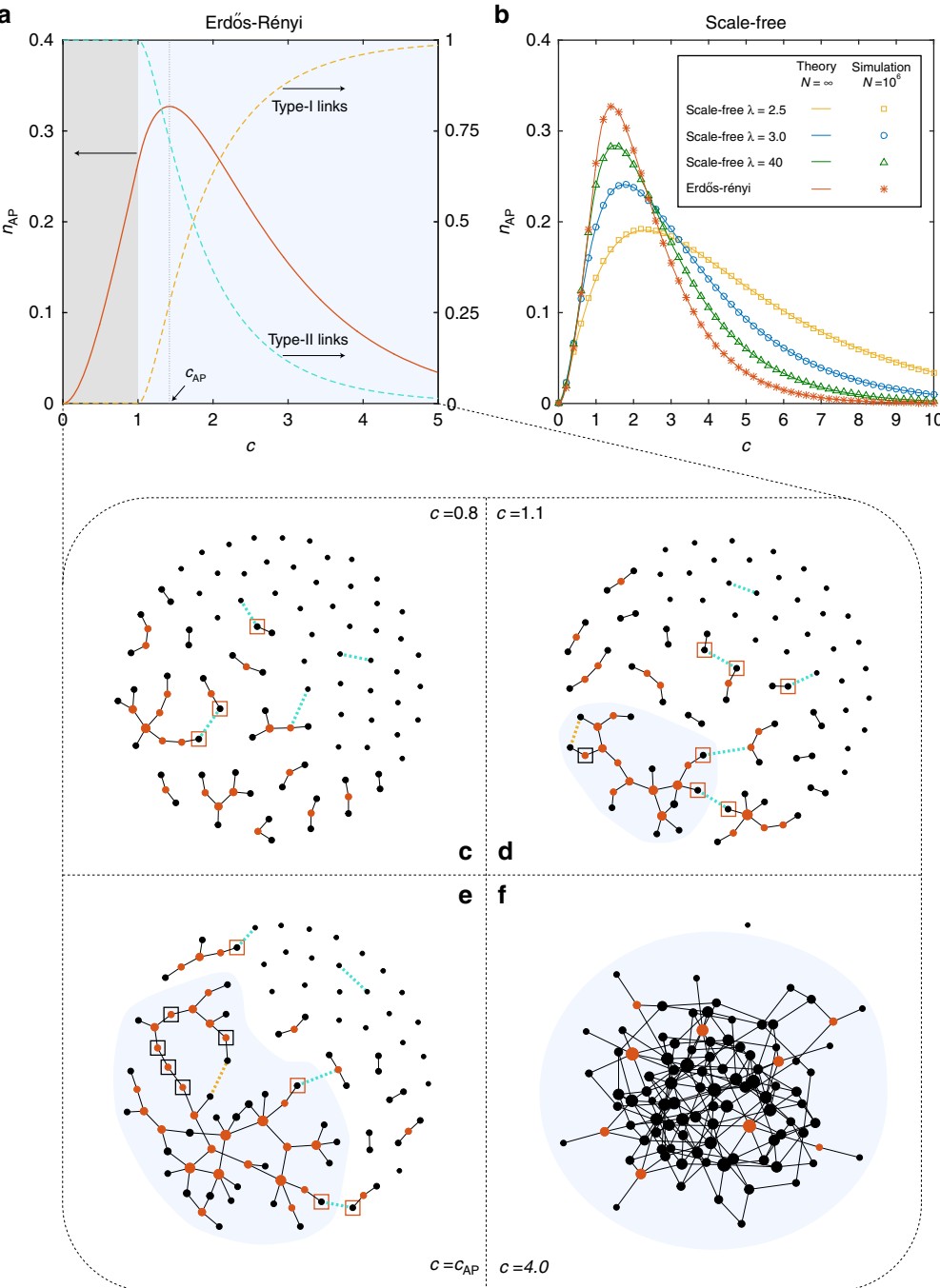

**Figure 3 | Fraction of articulation points in two canonical model networks.** (**a**) Erdős-Rényi random networks[20]; (**b**) Scale-free networks with different degree exponents $\lambda$. In **a**, the fraction of articulation points ($n_{AP}$) is shown as red line. The probabilities of adding type-I (yellow dashed line) and type-II links (turquoise dashed line) are also shown. In **b**, we use the static model[26] to construct scale-free networks with asymptotically power-law degree distribution $P(k) \sim k^{-\lambda}$. Simulations are performed with network size $N = 10^6$ and the results (symbols) are averaged over 128 realizations with error bars (s.e.m.) smaller than the symbols. Lines are our theoretical predictions. (**c–f**) Illustrations of articulation points (red nodes), type-I links (yellow dashed lines) and type-II links (turquoise dashed lines) in Erdős-Rényi random networks of different mean degrees. Note that adding a single type-II link at most convert two normal nodes to articulation points (orange boxes), while adding a single type-I link could convert much more articulation points back to normal nodes (black boxes). This explains why the peak of $n_{AP}$ emerges even though the probability of adding type-II links is still larger than that of adding type-I links. The largest connected component is highlighted in light blue in **d–f**.

critical singularity as a second-order phase transition. Interestingly, the GCC and RGB percolation transitions have completely different critical exponents associated with their critical singularities (Fig. 4c).

We also calculate the total number of GAPR steps $T(c)$ needed to remove all APs of an infinitely large ER network of mean degree $c$

(Fig. 4b). We find that $T(c)$ is finite for $c < c^\star$; diverges when $(c - c^\star) \to 0^-$; and is infinite for any $c > c^\star$. The divergence of $T(c)$ displays a scaling behaviour $T(c) \sim |c - c^\star|^{-\gamma_-}$ with critical exponent $\gamma_- = 1/2$ when $(c - c^\star) \to 0^-$ (Fig. 4d, magenta line).

The nature of the discontinuous RGB percolation transition and the behaviour of $T(c)$ can be revealed by analysing the

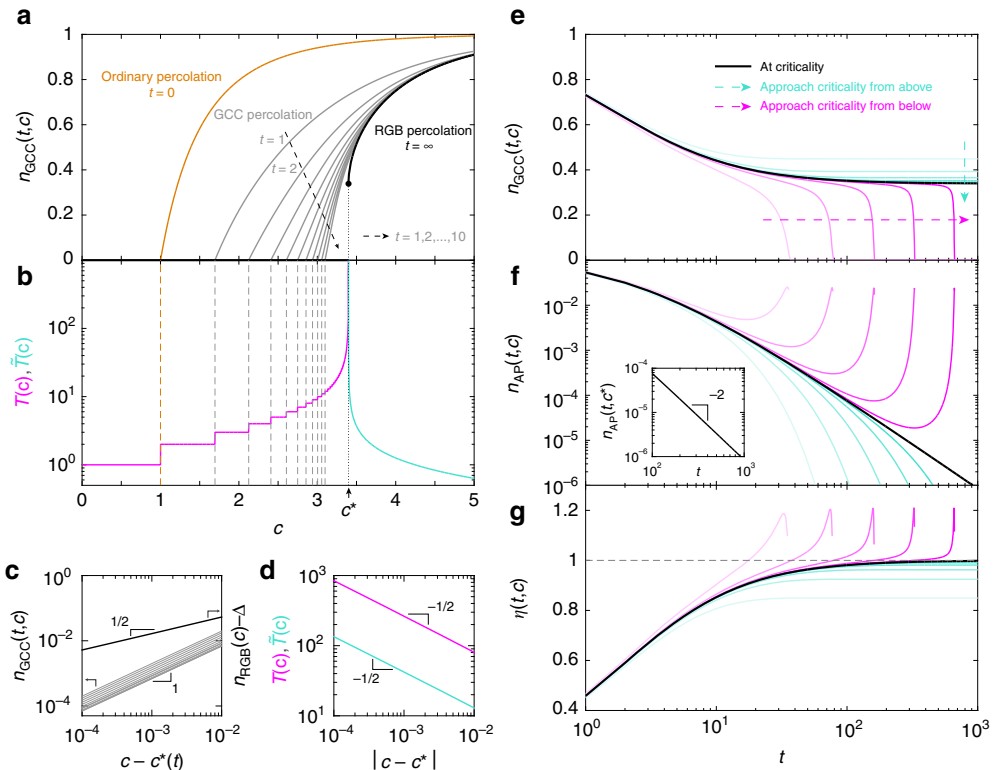

**Figure 4 | Percolation transitions associated with greedy articulation points removal.** Two types of percolation transitions of different nature are shown for Erdős-Rényi random networks. (**a**) Relative size of the giant connected component (GCC) after $t$ steps of greedy articulation points removal (GAPR), $n_{GCC}(t, c)$, as a function of the mean degree $c$. Note that $n_{GCC}(0, c)$ corresponds to the ordinary percolation (orange line); $n_{GCC}(t, c)$ with finite $t$ (only $t = 1$, 2,..., 10 are shown here) corresponds to the GCC percolation (grey lines); and $n_{GCC}(\infty, c) = n_{RGB}(c)$ corresponds to the residual giant bicomponent (RGB) percolation (thick black line). (**b**) Total number of the GAPR steps $T(c)$ for $c < c^\star$ (magenta line) and the characteristic number of the GAPR steps $\tilde{T}(c)$ for $c > c^\star$ (turquoise line) as functions of the mean degree $c$. (**c**) The critical scaling behaviour of $n_{GCC}(t, c)$ and $n_{RGB}(c)$ for the GCC and RGB percolation transitions, respectively. (**d**) The divergence of $T(c)$ and $\tilde{T}(c)$ associated with the RGB percolation transition. (**e–g**) Temporal behaviours of fraction of the GCC ($n_{GCC}(t, c)$), fraction of APs ($n_{AP}(t, c)$), and average number of newly induced APs per single AP removal $\eta(t, c)$ at critical (black lines), subcritical (magenta lines, $c - c^\star = -2^4 \times 10^{-5}$, $-2^6 \times 10^{-5}$, $-2^8 \times 10^{-5}$, $-2^{10} \times 10^{-5}$, $-2^{12} \times 10^{-5}$, respectively) and supercritical (turquoise lines, $c - c^\star = 2^4 \times 10^{-5}$, $2^6 \times 10^{-5}$, $2^8 \times 10^{-5}$, $2^{10} \times 10^{-5}$, $2^{12} \times 10^{-5}$, respectively) regions of the RGB percolation transition. At criticality, $n_{AP}(t, c^\star)$ decays in a power-law manner for large $t$ (inset of **f**).

dynamics of the GAPR process. In particular, we can calculate $n_{GCC}(t, c)$, $n_{AP}(t, c)$, as well as a key quantity in the GAPR process, that is, the average number of newly induced APs per single AP removal: $\eta(t, c) = n_{AP}(t, c)/n_{AP}(t - 1, c)$ for $t > 0$ and at different mean degrees $c$ (Fig. 4e–g).

For $c > c^\star$ (supercritical region), the fraction of APs exponentially decays as $n_{AP}(t, c) \sim \exp(-t/\tilde{T}(c))$ after an initial transient time (Fig. 4f, turquoise lines), where $\tilde{T}(c)$ is the characteristic time scale. In this region, with increasing $t$, $\eta(t, c)$ quickly reaches an equilibrium value $\eta(\infty, c) = \exp(-1/\tilde{T}(c))$, which is smaller than 1 (Fig. 4g, turquoise lines). Consequently, $n_{GCC}(t, c)$ converges to a finite value for $t \to \infty$ (Fig. 4e, turquoise lines), resulting in a finite $n_{RGB}$. Since $T(c)$ is infinite in this region, we can use $\tilde{T}(c)$ to characterize the relaxation behaviour of GAPR process. We find that $\tilde{T}(c)$ increases as $c$ decreases (Fig. 4b, turquoise line), and diverges as $\tilde{T}(c) \sim |c - c^\star|^{-\gamma_+}$ with critical exponent $\gamma_+ = 1/2$ when $(c - c^\star) \to 0^+$ (Fig. 4d, turquoise line). Note that as $c$ decreases and approaches $c^\star$ from above, the equilibrium value $\eta(\infty, c)$ gradually approaches 1 (Fig. 4g).

When $(c - c^\star) \to 0^+$ (that is, right above the criticality), the fraction of APs decays in a power-law manner for large $t$, that is, $n_{AP} \sim t^{-z}$ with $z = 2$ (Fig. 4f and inset, black line), rendering $\eta(\infty, c) = 1$ (Fig. 4g, black line). Consequently, $n_{GCC}(t, c)$ converges to a finite value in the $t \to \infty$ limit (Fig. 4e, black lines),

leading to a finite $n_{RGB}$. The fact $\eta(\infty, c) = 1$ suggests that in average every removed AP will induce one new AP at the next time step, and hence the GAPR process will continue forever. This explains why $T(c^\star)$ diverges. Note that the equilibrium value $\eta(\infty, c) = 1$ can only be reached when $n_{AP}(t, c)$ displays a power-law decay as $t \to \infty$. This is because, as long as $n_{AP}(t, c)$ is finite at any finite $t$, the GAPR will gradually dilute the network, rendering a larger and larger value of $\eta(t, c)$ as $t$ grows.

When $(c - c^\star) \to 0^-$ (that is, right below the criticality), as well as in the entire subcritical region ($c < c^\star$), after an initial decay, $n_{AP}(t, c)$ begins to exponentially grow with increasing $t$ (Fig. 4f, magenta lines). Consequently, $\eta(t, c)$ is initially < 1, but then becomes drastically larger than 1 (Fig. 4g, magenta lines), which causes $n_{GCC}(t, c)$ quickly decays to zero (Fig. 4e, magenta lines), and hence $T(c)$ is finite and the RGB dose not exist. The sudden collapse of the RGB upon an infinitesimal decrease in $c$ suggests the discontinuous nature of the RGB percolation transition in ER networks. Note that at time step $T$, the network will break into pieces and there is no AP left. Hence in the last few GAPR steps the growth of $n_{AP}(t, c)$ will slow down and eventually decrease (Fig. 4f,g, tails in the magenta lines).

For finite-sized networks sampled from a network ensemble with a prescribed degree distribution, the value of $n_{RGB}$ at criticality $c^\star$ is subject to large sample-to-sample fluctuations, being either zero or

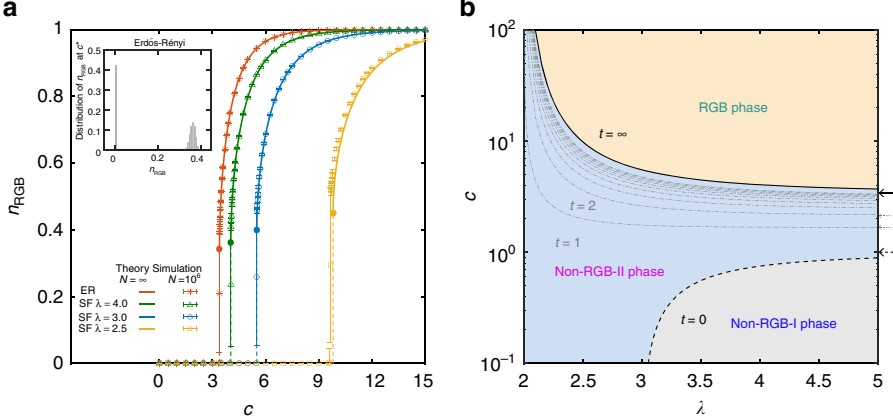

**Figure 5 | Residual giant bicomponent percolation transition and the phase diagram.** (**a**) Relative size of the residual giant bicomponent, $n_{RGB}$, as a function of the mean degree $c$ in the Erdős-Rényi network (red), and scale-free networks with different degree exponents, $\lambda = 4.0$ (green), 3.0 (blue) and 2.5 (yellow), constructed from the static model[26]. Lines are our theoretical predictions. Simulations are performed with network size $N = 10^6$. Results (symbols) are averaged over 128 realizations and the error bars (s.e.m.) are generally smaller than the symbols, except at criticality. The deviation of simulation results from our theoretical prediction for $\lambda = 2.5$ owes to degree correlations present in the constructed networks[46], which become prominent as $\lambda \to 2$. Inset displays the distribution of $n_{RGB}$ at criticality generated from 51,200 Erdős-Rényi networks of size $N = 10^6$. The bimodal distribution of $n_{RGB}$ indicates that it undergoes a discontinuous jump from nearly zero to a large finite value at the critical point. (**b**) Phase diagram associated with the greedy articulations point removal process in scale-free networks. The residual giant bicomponent percolation transition, the giant connected component percolation transitions (only $t = 1, 2,..., 10$ are shown here), and the ordinary percolation transition are shown in thick solid line, thin dot-dashed lines and thick dashed line, respectively. In the limit of large $\lambda$, the phase boundaries, $c^\star(t)$, for Erdős-Rényi networks are recovered (indicated by arrows). Here we only show $c^\star(t = \infty)$ (thick solid arrow), $c^\star(t = 1)$ and $c^\star(t = 2)$ (thin dot-dashed arrows), and $c^\star(t = 0)$ (thick dashed arrow).

a large finite value (Fig. 5a and inset), which is another evidence of discontinuous phase transition[29]. This discontinuous phase transition also partially explains the fact that real-world networks have either a very small or a rather big RGB (Fig. 2a).

The nature of the RGB percolation transition in SF networks is qualitatively the same as that in ER networks. The transition from the non-RGB phase to the RGB phase is discontinuous (Fig. 5a). The critical point $c^\star(\lambda)$ increases with decreasing $\lambda$. Also, the jump of the RGB relative size at criticality increases as $\lambda$ approaches 2 (Fig. 5a).

The $c - \lambda$ phase diagram of SF networks is shown in Fig. 5b. The whole diagram consists of three phases. For $c < c_p(\lambda)$ (grey region), there exists no GCC in the network, and hence no RGB. For $c_p(\lambda) < c < c^\star(\lambda)$ (light blue region), even though the GCC may survive after certain finite steps of GAPR, the RGB still does not exist. Since in both regimes there is no RGB, we call them non-RGB-I phase and non-RGB-II phase, respectively. The transition between these two phases is the ordinary GCC percolation transition, which is continuous (thick dashed line). Note that, in non-RGB-II phase, the phase transition associated with the emergence the GCC after any finite $t$ steps of GAPR is still continuous (thin dot-dashed lines). For $c > c^\star(\lambda)$ (light yellow region), the network suddenly has an RGB. This regime is referred to as the RGB phase. As we mentioned above, the transition between the non-RGB-II phase and the RGB phase is discontinuous (thick solid line). We have performed extensive numerical simulations to confirm our analytical results (Supplementary Note 6; Supplementary Figs 4–6).

Structural transitions in complex networks have been extensively studied and found to affect many network properties[10–12,14,29–43]. Here we show for the first time that there exists two different types of percolation transitions associated with the removal of APs.

## Discussion

In this article, we systematically investigate AP-related issues in complex networks. Many interesting phenomena of APs are

discovered and explained for the first time. On the empirical side, we proposed two AP-based applications: a network attack strategy (APTA) and a network decomposition method (GAPR). We found that, given a limited 'budget' (that is, the number of nodes to be removed), our APTA strategy is more efficient in reducing the GCC of the network than other existing strategies. In revealing the core-periphery structure of complex networks, our GAPR method is quite different from traditional network decomposition methods in the sense that our identified core may include low-degree nodes. Those sparsely connected nodes can be functionally very important, but they are always ignored in traditional decomposition methods. On the theoretical side, we proposed an analytical framework to calculate various AP-related properties, among which the emergence of the RGB as a discontinuous percolation transition is of great theoretical interest. This finding also provides a theoretical explanation of the empirical findings that most of the real-world networks have either a very small RGB or a rather big one.

Taken together, our results offer a different perspective on the organizational principles of complex networks, shed light on the design of more resilient infrastructure networks and more effective destructions of malicious networks, and open new avenues to deepening our understanding of complex networked systems. Since the identification of APs also helps us better solve other challenging problems, for example, the calculation of determinants of large matrices[44], and the minimum vertex cover problem on large graphs (a classical NP-hard problem)[45], we anticipate that our results on APs will trigger more research activities on those problems as well.

## Methods

**Theoretical analysis of greedy articulation points removal process.** Our theoretical treatment of the GAPR process is based on the local tree approximation, which assumes in the thermodynamics limit (that is, network size $N \to \infty$) there are no finite loops in a network and only infinite loops exist[23–25,28]. This approximation allows us to use the convenient techniques of random branching processes to solve the GAPR process on large uncorrelated random networks (Supplementary Note 4). Note that the local tree approximation is only exact for networks with finite second moment of the degree distribution. However, it has

been demonstrated in various network problems that this approximation can obtain very accurate results even for networks with diverging second moment of the degree distribution[28]. Here we find that this local tree approximation works very well in analysis of the GAPR process (Supplementary Notes 4 and 6).

At each time step $t$ during the GAPR process in a network $\mathcal{G}$, we classify the remaining nodes into the following three categories or states: (1) $\alpha_t$-nodes: nodes in FCCs; (2) $\beta_t$-nodes: nodes that are APs in the GCC; (3) $\gamma_t$-nodes: nodes that are not APs in the GCC. Note that if a node is a $\gamma_t$-node, it must be $\gamma_\tau$-node with $\tau < t$. (The notations $\beta_t$ and $\gamma_t$ here have totally different meanings from the critical exponents $\beta$ and $\gamma$ mentioned in the main text.)

According to the local tree approximation (Supplementary Note 4), the state of a randomly chosen node $i$ can be determined by the states of its neighbours in $\mathcal{G} \setminus i$, that is, the induced subgraph of $\mathcal{G}$ with node $i$ and all its links removed. In other words, in order to determine the state of a node, we need to know the states of its neighbours. Therefore, at each time step $t$, we need to know the probability that, following a randomly chosen link to one of its end nodes, this node belongs to any of the above categories after this link is removed. These probabilities are denoted as $\alpha_t$, $\beta_t$, and $\gamma_t$, respectively. Note that for convenience sake here we use the same notation to denote both the state of a node and the probability of a node in that state. To be precise, hereafter when we consider the state of a neighbour of a given node $i$, we mean the state of the neighbour in the induced subgraph $\mathcal{G} \setminus i$.

The GAPR process can be fully characterized by the three sets of probabilities $\{\alpha_0, \alpha_1,...\}$, $\{\beta_0, \beta_1,...\}$ and $\{\gamma_0, \gamma_1,...\}$. Note that every node must belong to one of the three categories, which means the three sets of probabilities are not independent from each other. Specifically, at time step $t$, the probability $\gamma_t$ can be derived by the other two sets of probabilities through the following normalization condition:

$$\sum_{\tau=0}^{t} \alpha_\tau + \sum_{\tau=0}^{t} \beta_\tau + \gamma_t = 1. \qquad (1)$$

Hereafter we focus on $\alpha_t$ and $\beta_t$ only. We can calculate $\{\alpha_0, \alpha_1,...\}$ and $\{\beta_0, \beta_1,...\}$ in an iterative way. At first, we consider the initial time step $t = 0$. The self-consistent equations for $\alpha_0$ and $\beta_0$ are given by

$$\alpha_0 = \sum_{k=1}^{\infty} Q(k)(\alpha_0)^{k-1} \qquad (2)$$

$$\beta_0 = \sum_{k=3}^{\infty} Q(k) \left[ 1 - (1-\alpha_0)^{k-1} - (\alpha_0)^{k-1} \right], \qquad (3)$$

where $Q(k) = kP(k)/c$ is the degree distribution of the nodes that we arrive at by following a randomly chosen link (a.k.a. the excess degree distribution)[1,2]. We derive the above equations based on the following observations: (1) $\alpha_0$-node: its neighbours can only be $\alpha_0$-nodes; (2) $\beta_0$-node: since it is an AP node, at least one of its neighbours is an $\alpha_0$-node. Moreover, since it belongs to the GCC, at lease one of its neighbours is not an $\alpha_0$-node.

For the $t$-th GAPR time step ($t > 0$), we can compute $\alpha_t$ and $\beta_t$ as follows:

$$\alpha_t = \sum_{k=1}^{\infty} Q(k) \left( \alpha_t + \sum_{\tau=0}^{t-1} \beta_\tau \right)^{k-1} - \sum_{k=1}^{\infty} Q(k) \left( \sum_{\tau=0}^{t-2} \beta_\tau \right)^{k-1} \qquad (4)$$

$$\beta_t = \sum_{k=3}^{\infty} Q(k) \sum_{s=1}^{k-2} \binom{k-1}{s} \left( 1 - \sum_{\tau=0}^{t} \alpha_\tau - \sum_{\tau=0}^{t-1} \beta_\tau \right)^s \times \sum_{r=1}^{k-1-s} \binom{k-1-s}{r} (\alpha_t)^r \left( \sum_{\tau=0}^{t-1} \beta_\tau \right)^{k-1-s-r}. \qquad (5)$$

The derivations of equations (4 and 5) are based on the following observations: (1) $\alpha_t$-node: First, its neighbours can only be $\alpha_t$-nodes or $\beta_\tau$-nodes with $\tau < t$ (because if one of its neighbours is $\alpha_\tau$-node with $\tau < t$, this node will be an AP before time step $t$ and hence would have already been removed; if one of its neighbour is $\beta_t$-node, this node will belong to the GCC at time step $t$). Second, its neighbours can not be all $\beta_\tau$-nodes with $\tau < t-1$. Otherwise this node will be a leaf node before time step $t-1$, and will become an isolated node before the $t$-th time step. In this case, we can not reach this node through a randomly chosen link at time step $t$. (2) $\beta_t$-node: First, its neighbours can not be $\alpha_\tau$-nodes with $\tau < t$, otherwise this node would have been removed before time step $t$. Second, since it is an AP node, at least one of its neighbours is an $\alpha_t$-node. Finally, since this node belongs to the GCC, at least one of its neighbours is neither $\alpha_t$-node nor $\beta_\tau$-node with $\tau < t$.

Diagrammatic representations of these probabilities ($\alpha_t$, $\beta_t$, $\gamma_t$) and their relationship are shown in Supplementary Figs 7–9 (see Supplementary Note 4 for details).

By solving the above self-consistent equations, we can obtain $\{\alpha_0, \alpha_1,...\}$ and $\{\beta_0, \beta_1,...\}$, which govern the whole process of GAPR. With these two sets of probabilities, we can compute any quantities of interest, such as the total number of GAPR steps, the fraction of APs, the relative size of the GCC and the RGB, and so on (Supplementary Note 4; Supplementary Figs 10–12).

**Data availability.** The data that support the findings of this study are available from the corresponding author on reasonable request.

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

## Acknowledgements

We thank Shlomo Havlin, Mehran Kardar, Wei Chen, Endre Csóka, Abhijeet Sonawane, Yandong Xiao, Chuliang Song for valuable discussions. This work was partially supported by the John Templeton Foundation (Award No 51977), National Natural Science Foundation of China (grants nos. 11505095 and 11374159), and the Fundamental Research Funds for the Central Universities of China (grants nos NS2014072 and NZ2015110).

## Author contributions

Y.-Y.L. conceived and designed the project. L.T. performed all the analytical calculations and extensive numerical simulations, as well as empirical data analysis. All authors analysed the results. L.T. and Y.-Y.L. wrote the manuscript. A.B. and D.-N.S. edited the manuscript.
