## [Peer Review File · Nature Communications]

Reviewer #1 (Remarks to the Author):

The paper presents a novel way to analyze large networks in terms of their articulation points and sizes of connected components that remain after their successive removal. To do that, a novel network decomposition method is introduced. In addition, analytical results are provided on the dynamics of such a decomposition for Erdos-Renyi and Scale-free networks. Two percolation transitions of different nature have been found. These results present significant contributions to network science with a potential to impact several real-world domains that use network science as a tool to answer some fundamental natural or societal questions from networked data.

A drawback of the paper is that it provides a limited link of the introduced method and the results to real-world networks. Except for a brief discussion and a description of how the decomposition works in the 9/11 terrorist network at the beginning of the paper, after the elaborate analytical framework, there was no application to the real world data. I would have hoped to see how the presented method and analytical results contribute to deepening of our understanding of at least one large current real-world network.

Overall, the methodology is novel and interesting with a high potential to contribute solving some problems involving real-world network data.

Reviewer #2 (Remarks to the Author):

Articulation points (Aps) are nodes that, if removed, the number of connected components in the network increases. Identify the APs and understand their properties are questions of vital importance for network resilience (e.g., infrastructural networks) and mitigation of dark networks (e.g., criminal networks). In this article, the authors propose a theoretical framework to study APs of uncorrelated (random) networks. With this new framework, they identify a new discontinuous percolation transition. Direct numerical simulations confirm all analytical results.

This new framework is definitely relevant and the results interesting. But, to learn about the most important part of the work one need to read the supplemental information. As recognized by the authors in the introduction and conclusion, the main message is the new framework. However, in the main text, we cannot even find a brief description of its main ideas. I suggest that the authors write a completely new session of the paper devoted to a brief description of the framework. This section should focus on the key ideas and leave the technicalities to the supplemental information.

The idea of an RGB resembles the core of the triangle decomposition (t-core decomposition). The authors should discuss what is the relation (if any) between the RGB and the core of a core-periphery structure.

Reviewer #3 (Remarks to the Author):

Review for "Articulation Points in Complex Networks" by Liang Tan, Amir Bashan, Da-Ning Shi, and Yang-Yu Liu

SUMMARY OF CONTENT

The authors study the problem of targeted attacks on the connectivity of complex networks by removing specific nodes, i.e. articulation points in this case. As they focus mainly on the size of the giant connected component (GCC), the problem can be interpreted as variant of percolation. Novel is their definition of two algorithms for the stepwise removal of articulation points, which allow the identification of some kind of core-ness of nodes with respect to the proposed procedures.

The first algorithm is a Greedy removal of the articulation point causing the highest damage (APTA). In contrast, the second removes all identified articulation points in a time step (GAPR). While the first one is compared with other targeted attacks and shows more effective results in some cases, the second one can be described analytically on random configuration model type random graphs.

The main argument to present both analyses together is that the two algorithms lead to similar (overlapping) sets of articulation points that are removed and consequently to a similar residual giant bicomponent (RGB).

GENERAL POINTS OF CRITIQUE

The studied problems are quite interesting.

However, at this stage, the amount of material seems to be overwhelming and is presented in a partially confusing distribution to the main paper and supplementary material. The text could be significantly shortened and the core message sharpened.

For instance, already improving the introduction and conclusions could support the readability significantly.

Introduction:

The results for the terrorist network and its description don't need mentioning early on in this detail. Also the study of articulation points and relevance is quite intuitive and does not need to be motivated that extensively.

A topic sentence in the end of the introduction instead could help understand the general line of argument. From my point of view, there are two big main topics that should be mentioned. (Alternatively, they could also be easily split in two papers.)

First, there is the APTA algorithm, which is presented as very effective targeted attack to the connectivity of a network. This is shown in comparison to other known attack options on several real world networks and on random graphs. Furthermore, structural properties about a network are revealed by the time of removal/coreness of specific nodes. One detailed example is then provided by the terrorist communication network. The empirical findings of either rather big or small RGBs motivates to study the process closer on (configuration model) random graphs.

As this study shall be studied analytically, the APTA algorithm is modified so that it can be treated by a local tree approximation/branching process approximation/heterogeneous mean field approximation. (Hint: You might like to use these terms, as they are used quite often.)

(The main advantage is, by the way, not so much that it is deterministic. Most importantly, it gets rid of optimizing the damage caused by a removed node.)

The second main topic is the analysis of this alternative GAPR algorithm that still leads to similar results as APTA in the end. It is very nice that numerically precise derivations of the results are provided. Personally to me, this is one of the main contributions of the paper.

However, it is important to mention at least later in the main text: The analytic derivation applies only to infinitely large networks (in the thermodynamic limit) and only to degree distributions with finite second moment.

Results:

A more dense section where all definitions and abbreviations are introduced next to each other could serve as reference to return to in the course of reading. The amount of abbreviations is quite substantial and it is easy to lose track.

Conclusions:

The complex network history and general philosophy could be substantially shortened. Instead, a more detailed summary of the findings and results would be appropriate and would orient a confused reader. What are the actual contributions?
What is the line of thought again? Why are APTA and GAPR studied? Which phase transitions occur? Why are they relevant and why do the findings explain the empirical observations obtained with APTA?

DETAILED POINTS OF CRITIQUE

The abbreviation p. means page, while SI refers to Supplementary Information.

Main paper:

- p.1: The analytic framework that you develop does not apply to an arbitrary complex network. You can calculate the average mentioned quantities for configuration model type random graphs where the prescribed degree distribution has finite second moment.
- p.1: What is a "rich" phase diagram?! You could simply delete this part.
- p.2: Please comment that single isolated nodes do not count as component to you (or otherwise if I have not understood correctly).
- p.2: High potential for shortening. For instance, the sentence "Those AP nodes play an important role in ensuring the connectivity..." is not needed. Its implied by their definition.
- p.2: Not nice formulation: "notorious" NP-complete problem
- p.3: It would be helpful to mention with what kind of attacks to compare APTA with in the SI.
- p.3: "Structural integrity" sounds strange to me.
- p.3: Please, clarify in the paper: How does GAPR precisely work? If two APs are connected so that the removal of one would be enough to split the network in two components, do you still remove both APs? I guess so from the text.
- p.4: "still lack a deep understanding on the roles..." -> "still lack a deep understanding of the roles..."
- p.4 (Analysis of Real Networks): "relative size": Please, add relative to the total number of nodes in the network
- p.4 (Analysis of Real Networks): "relative size": I personally find the notation $n_{\{*\}}$ a bit unfortunate, because I would think you refer to a number. What about r or ρ or d ?
- p.5: "Ironically" sounds strange to me in a paper.
- p.5: Comment: It's not surprising to find the mentioned result. The studied (infrastructure) networks are not optimized with respect to APTA, of course. They basically lack a high redundancy, because building a link comes with a high cost. (Still, the networks are often optimized with respect to other criteria that make sense.) Of course, you can suggest to take your considerations into account to build more robust systems.
However, you would need to argue why you only care about the GCC and not the size of the other components. Maybe, some of your networks are able to maintain a certain functionality without being connected to the largest component. Or, also the largest component cannot function without the rest, because a critical node was removed. The functionality of each node and how they contribute to the whole functionality would decide about the answers to these questions.
Still, you present a nice proxy and the motivation to study GAPR processes is clear.
- p.5: In what respect is the phase transition "novel"?
- p.5: Not all deterministic processes can be described analytically. What is really important to make GAPR tractable is that you get rid of the optimization of damage that an AP causes in APTA.
- p.6: The model with prescribed degree sequence/degree distribution is called configuration model

(as you name it later) and goes back to:

```
@article{Molloy1995,  
author = {Molloy, Michael and Reed, Bruce},  
journal = {Random structures {\&} algorithms},  
number = {1995},  
pages = {161--179},  
title = {{A critical point for random graphs with a given degree sequence}},  
volume = {6},  
year = {1995}  
}
```

-p.6: Some of your networks are small enough that the simulation results lead to rather broad distributions. This means that not all networks with the given prescribed degree distribution have the same properties and some in the sample also have degree-degree correlations, for instance. Thus, you cannot simply attribute the properties that you observe to the degree distribution in all of these cases.

-p.6 (Results): What about a transitional sentence that now you move the attention now to infinitely large networks and a more in depth exploration with respect to different degree sequences?

-p.7: State your assumptions more clearly: Infinitely large networks, finite second moment of degree distribution required, in which order do you take the thermodynamic limit and $c \rightarrow \infty$?

-p.7: $\lambda > 3$ or you need a finite cut-off degree.

-p.7: c is not generally defined as average degree. It is just introduced as parameter of the Poisson degree distribution.

-p.7: In general, you do not need to give so many digits of c_{AP} (or other numerical values). Two are fully enough.

-p.7: Please, cite a supporting article for $c_p = 1$ for percolation in ER networks.

-p.8: Does your argument for ER networks with adding links also apply to SF networks? Isn't it important which nodes with which degree receive higher degrees? Refer also to SI for explanation how the increase of average degree works.

-p.8: Add one step in the argument in the second paragraph: For SG networks: GCC's relative size is rather small..., - and the network is more fragmented in small components - which results in a larger c_{AP} .

-> Why does it really need to? Doesn't it depend on how the components are connected among each other and not their size or number? For instance, many (small or large) fully connected cliques do not have any APs. It really needs to have something to do with the high degree heterogeneity.

-p.9: In general not a too nice formulation in a paper: "which is nothing but" -> i.e.

-p.9 or 10: Another topic sentence as introduction would be nice. A small summary of what you are going do present and why you analyze the specific variables that you analyze.

-p.13: The results are not discussed at all! This would be a good opportunity to forge a bridge to the introduction and discuss the implications of your work.

-Graphics:

Considering the space that you use for all the plots, you could make them much larger and better readable. Some color drawings almost vanish when you print the paper.

Define: abbreviation s.e.m.

Supplementary information:

-p.4: The name configuration model could be mentioned again.

-p.4/6: Please, clarify the GAPR: Do you remove APs that are connected to each other? (I guess that yes.)

-p.7: Not everyone knows what a thermodynamic limit is. Please explain that you study the limit $N \rightarrow \infty$.

-p.7: The method that you use is also called Local Tree Approximation/ Branching Process

Approximation/ Heterogeneous Mean Field Approximation. It's not really a tree ansatz, because you only consider locally tree-like structures, not real trees. This makes a huge difference. In a tree, every node with degree > 1 would be an AP.

-p.7: Important: The locally tree like structure only appears for finite second moment of the degree distribution!

This also explains why your numerical calculations do not match perfectly the simulations for $\lambda \leq 3$.

p.7: Last paragraph in "Tree ansatz": See also:

```
@article{Melnik2011,  
author = {Melnik, Sergey and Hackett, Adam and Porter, Mason a. and Mucha, Peter J and  
Gleeson, James P},  
doi = {10.1103/PhysRevE.83.036112},  
issn = {1539-3755},  
journal = {Physical Review E},  
number = {3},  
pages = {36112},  
title = {{The unreasonable effectiveness of tree-based theory for networks with clustering}},  
volume = {83},  
year = {2011}  
}
```

p.7: Definition of type of nodes together with Eq. (2) on p.8: I did not understand what you mean from the text that you write on p.7. On the basis of Eq. (2). I guess, you mean: β_t -nodes: nodes that are identified as AP nodes exactly at time t and belong to the GCC at time t .

γ_t -nodes: nodes that belong to the GCC all the times $\{0, \dots, t\}$.

State explicitly that α_t also includes APs.

p.7: β_t and γ_t are not defined as probabilities on p.6, when you define $\beta_t + \gamma_t$!

p.8: A sentence would help why you focus on probabilities of the end node type of a link first. The goal is to calculate n_{AP} and N_{RGB} etc. There, you need to know the type of neighbors of a node to judge the state of a node. Thus, you calculate the probability first that neighbors of a node (i.e. the end nodes of a randomly drawn link) are of a certain type.

-p.9: Fig.8 is pretty trivial and not needed from my point of view. The introduction of notation could be added to the next figure.

-p.9: Tree ansatz not good formulation.

-p.9: It is absolutely crucial for your derivations that the removal of an AP does not break down the GCC in two components of non-zero relative size. This needs more explanation or at least a reference for a proof.

-p.11: Technically, $G_1(x)$ is not the generating function of $Q(k)$. This would be $\sum_k Q(k)x^k$. $G_1(x) = G'_0(x)/c = G'_0(x)/G'_0(1)$, where $G_0(x) = \sum_k P(k)x^k$.

-p.12: State more clearly: ...: Fraction of APs whose removal splits the GCC in a smaller GCC and FCCs

...: Fraction of APs whose removal splits an FCC in a smaller FCCs.

-p.14: State why you write the formulas in terms of generating functions... Eases calculations later, when you have explicit close form solutions for G_0 .

-p.17: Definition of a : You mean $a = 1/(\lambda - 1)$, I guess.

-p.20 and following: As written again. Consider that your analytic derivations only hold for degree distributions with finite second moment. The cases of SF with $\lambda = 3$ and $\lambda = 2.5$ do not need to work theoretically.

Overall, the work is very interesting and a deep analysis is provided. However, considering the high amount of material, it needs additional effort to present it in an more easily accessible way.

Yours sincerely,
Rebekka Burkholz

Response to Reviewer #1

The paper presents a novel way to analyze large networks in terms of their articulation points and sizes of connected components that remain after their successive removal. To do that, a novel network decomposition method is introduced. In addition, analytical results are provided on the dynamics of such a decomposition for Erdos-Renyi and Scale-free networks. Two percolation transitions of different nature have been found. These results present significant contributions to network science with a potential to impact several real-world domains that use network science as a tool to answer some fundamental natural or societal questions from networked data.

We thank Reviewer #1 for his/her very positive assessment of the novelty, general interest and fundamental importance of our work.

A drawback of the paper is that it provides a limited link of the introduced method and the results to real-world networks. Except for a brief discussion and a description of how the decomposition works in the 9/11 terrorist network at the beginning of the paper, after the elaborate analytical framework, there was no application to the real world data. I would have hoped to see how the presented method and analytical results contribute to deepening of our understanding of at least one large current real-world network.

We thank Reviewer #1 for this very insightful comment.

In the beginning of our paper, we described the detailed AP-based decomposition process for the 9/11 terrorist network as an example to demonstrate the definitions of AP and the residual giant bicomponent (RGB), as well as the potential functionality of the RGB. This network was chosen for two reasons: (1) It is small enough for clear visualization purpose; (2) It is well documented and labeled so that we can identify the role of each node.

In principle, our AP-based decomposition method can be applied to any real-world network. And the functionality of the RGB can be systematically studied if the network is well labeled (in the sense that we know the attributes of the nodes and edges). Unfortunately, not all the real-world networks accessible to us are well labeled. For many of them, we only have the edge list with nodes simply labeled by integers. The detailed node attributes are completely unavailable to us, rendering the functionality analysis infeasible.

Despite the limitation of currently available real data, to fully demonstrate the real-world applications of our framework, we have made the following efforts in our manuscript:

1. We calculated the fraction of APs and the relative size of RGB for 14 different types of real-world networks (from technological networks to infrastructure networks, biological networks, communication networks and social networks, etc.). Our results were presented in Fig. 2 of the main text. Even without the functionality analysis, we still found some interesting results. For example, we found that most of the real networks have a non-ignorable fraction of APs; and either no RGB or a rather large one. Surprisingly, we found that infrastructure networks (such as power grids and road networks) have relatively large fraction of APs and a small RGB, and hence are much more vulnerable

than we would expect. By contrast, most of food webs we analyzed have no APs, indicating that the extinction of one species will not disconnect the whole network at all.

2. We performed the AP-targeted attack (APTA) for several real-world networks, finding that, for a small fraction of removed nodes, APTA leads to the quickest reduction of the giant connected component (GCC) for those networks (see Supplementary Fig. 1 in the Supplementary Information). This result clearly demonstrates the fundamental importance of AP nodes in maintaining the connectivity of real-world networks. Indeed, APs are natural targets of attack if one aims for immediate damage to a network.

We think those findings already deepen of our understanding of many large real-world networks. Of course, with more node attributes available, we can perform more systematic functionality analysis of those APs, as well as the RGB nodes. We hope this current work will trigger a burst of research activities in the promising direction.

Overall, the methodology is novel and interesting with a high potential to contribute solving some problems involving real-world network data.

We thank Reviewer #1 again for his/her very insightful and constructive comments. We hope our response above have addressed those very legitimate issues/concerns in a satisfactory manner.

Response to Reviewer #2

Articulation points (Aps) are nodes that, if removed, the number of connected components in the network increases. Identify the APs and understand their properties are questions of vital importance for network resilience (e.g., infrastructural networks) and mitigation of dark networks (e.g., criminal networks). In this article, the authors propose a theoretical framework to study APs of uncorrelated (random) networks. With this new framework, they identify a new discontinuous percolation transition. Direct numerical simulations confirm all analytical results. This new framework is definitely relevant and the results interesting.

We thank Reviewer #2 for his/her very positive assessment of the novelty, technical soundness, relevance and general interest of our work.

But, to learn about the most important part of the work one need to read the supplemental information. As recognized by the authors in the introduction and conclusion, the main message is the new framework. However, in the main text, we cannot even find a brief description of its main ideas. I suggest that the authors write a completely new session of the paper devoted to a brief description of the framework. This section should focus on the key ideas and leave the technicalities to the supplemental information.

We thank Reviewer #2 for this excellent suggestion. In the revised manuscript, we added a completely new session entitled “Methods” (see main text, page 15) devoted to a brief description on the key ideas of our framework.

The idea of an RGB resembles the core of the triangle decomposition (t-core decomposition). The authors should discuss what is the relation (if any) between the RGB and the core of a core-periphery structure.

We thank Referee #2 for this excellent suggestion. In the revised SI (Page 30, Supplementary Note 3), we added a new section entitled “Core-periphery structure” to discuss the relations and differences between RGB and other cores identified through various traditional methods, e.g. maximum clique, k -core decomposition, and t -core decomposition. We also added a brief description of the difference between GAPR and those traditional methods in the revised main text (page 4, 1st paragraph).

Basically, all these methods aim to identify certain core-periphery structure of a network. Traditional methods are designed to uncover or extract a dense core structure within a network, i.e., a subnetwork consisting of *highly connected* nodes. However, the RGB identified through GAPR may also include *sparsely connected* nodes, which can be functionally very important but are always ignored in traditional methods. Hence, our AP-based decomposition method provides us a new angle to analyze and understand the organizational principles of real-world complex networks.

Finally, we thank Reviewer #2 again for his/her very insightful and constructive comments. We hope our response above have addressed those very legitimate issues/concerns in a satisfactory manner.

Response to Reviewer #3

Review for "Articulation Points in Complex Networks" by Liang Tan, Amir Bashan, Da-Ning Shi, and Yang-Yu Liu

SUMMARY OF CONTENT

The authors study the problem of targeted attacks on the connectivity of complex networks by removing specific nodes, i.e. articulation points in this case. As they focus mainly on the size of the giant connected component (GCC), the problem can be interpreted as variant of percolation. Novel is their definition of two algorithms for the stepwise removal of articulation points, which allow the identification of some kind of coreness of nodes with respect to the proposed procedures. The first algorithm is a Greedy removal of the articulation point causing the highest damage (APTA). In contrast, the second removes all identified articulation points in a time step (GAPR). While the first one is compared with other targeted attacks and shows more effective results in some cases, the second one can be described analytically on random configuration model type random graphs. The main argument to present both analyses together is that the two algorithms lead to similar (overlapping) sets of articulation points that are removed and consequently to a similar residual giant bicomponent (RGB).

GENERAL POINTS OF CRITIQUE

The studied problems are quite interesting.

We thank Reviewer #3 for his/her positive assessment of the novelty and the general interest of our work. Next we address each point raised by the Reviewer in order.

However, at this stage, the amount of material seems to be overwhelming and is presented in a partially confusing distribution to the main paper and supplementary material. The text could be significantly shortened and the core message sharpened. For instance, already improving the introduction and conclusions could support the readability significantly.

We thank Reviewer #3 for this excellent suggestion. We apologize for not making our previous manuscript well presented. We have substantially improved the presentation in the revised manuscript by taking into account the comments of Reviewer #3. Please see below for the details.

Introduction:

The results for the terrorist network and its description don't need mentioning early on in this detail. Also the study of articulation points and relevance is quite intuitive and does not need to be motivated that extensively. A topic sentence in the end of the introduction instead could help understand the general line of argument.

We thank Reviewer #3 for those very constructive suggestions.

Even though the importance and motivation of studying APs are apparent, we would like to keep the discussions about the APs in several real-world networks. There are two reasons. First, these examples directly support our statement “**Those AP nodes play important roles in ensuring the robustness and connectivity of many real-world networks**” in this paragraph. Second, discussing these examples can make the studied problem better understood by readers who are not familiar with APs. It helps us maintain the broad interest of our paper.

Following the suggestion of Reviewer #3, we added a topic sentence to the end of the first paragraph, which reads as follows: “**Analysis of APs hence provides us a new angle to systematically investigate the structure and function of real-world networks**”. We moved the original last sentence of the first paragraph (regarding the relationship between APs and large matrices determinants and vertex cover problem) to the discussion section. We feel it is not directly related to the main topics of our work, but could trigger more research activities on APs.

Also, according to the suggestion of Reviewer #3, we moved the description of the terrorist network from the beginning of the 2nd paragraph to the 3rd paragraph, where the AP-based network decomposition method was introduced.

From my point of view, there are two big main topics that should be mentioned. (Alternatively, they could also be easily split in two papers.)

We thank Reviewer #3 for this suggestion. We feel that the two big main topics are closely related to each other, and hence we really want to address them in one paper, using a unified framework. We do expect that this work will trigger a burst of research activities in this field.

First, there is the APTA algorithm, which is presented as very effective targeted attack to the connectivity of a network. This is shown in comparison to other known attack options on several real world networks and on random graphs. Furthermore, structural properties about a network are revealed by the time of removal/coreness of specific nodes. One detailed example is then provided by the terrorist communication network.

We thank Reviewer #3 for this suggestion. We think it is better to introduce the network attack algorithm (APTA) and the network decomposition method (GAPR) in two different paragraphs (2nd and 3rd paragraphs in the main text). This is simply because APTA and GAPR represent two different aspects/angles (the structure and the function, separately) to analyze networks, supporting the topic sentence in the end of the first paragraph. Also, both APTA and GAPR are completely new concepts, which we think deserve two paragraphs of descriptions.

As mentioned above, in the revised manuscript we have moved the description of the terrorist network from the beginning of the 2nd paragraph to the 3rd paragraph where GAPR is introduced.

The empirical findings of either rather big or small RGBs motivates to study the process closer on (configuration model) random graphs.

We thank Reviewer #3 for this suggestion. We consider the current logic flow of the paper (Introduction; Empirical Results; Analytical Results; Conclusion) is easier to follow. In this way, the finding that most of the real-world networks have either a small RGB or a rather big one is one of the main results in this work. Therefore, it is more appropriate to present this finding in the subsection “Analysis of Real Networks” of the section “Results”.

As this study shall be studied analytically, the APTA algorithm is modified so that it can be treated by a local tree approximation/branching process approximation/heterogeneous mean field approximation. (Hint: You might like to use these terms, as they are used quite often.)

We thank Reviewer #3 for this excellent suggestion. We would like to point out that term “tree ansatz” (or “tree approximation”) is often used in physics literature [1, page 1282], which is also known as the Bethe-Peierls approximation as widely used in spin glass theory and statistical inference [2]. In the revised manuscript, we did mention the other terms suggested by Reviewer #3 to make our paper more readable.

[1] S. N. Dorogovtsev, A. V. Goltsev, and J. F. F. Mendes, Critical phenomena in complex networks. *Reviews of Modern Physics* 80, 1275 (2008).

[2] M. Mézard and A. Montanari, *Information, Physics, and Computation* (Oxford Univ. Press, 2009).

(The main advantage is, by the way, not so much that it is deterministic. Most importantly, it gets rid of optimizing the damage caused by a removed node.)

We thank Reviewer #3 for pointing this out. We fully agree with Reviewer #3 that determinism is only one of the advantages of studying GAPR. Avoiding the optimization is the key reason why GAPR is theoretically tractable. In the revised manuscript, we have added the following sentence (see main text, page 5, 1st paragraph):

Note that, compared with APTA, the GAPR process is deterministic and avoids the optimization of the damage caused by nodes removal, which make it analytically solvable. Hereafter we focus on the RGB obtained from the GAPR process.

The second main topic is the analysis of this alternative GAPR algorithm that still leads to similar results as APTA in the end. It is very nice that numerically precise derivations of the results are provided. Personally to me, this is one of the main contributions of the paper.

We thank Reviewer #3 for his/her positive assessment of this finding. Similar as the finding that either a rather big or a small RGB exists in real networks, this result is also empirical. In order to

keep the structure of the paper, we feel it is better to present this finding in the subsection “Analysis of Real Networks” of the section “Results”.

However, it is important to mention at least later in the main text: The analytic derivation applies only to infinitely large networks (in the thermodynamic limit) and only to degree distributions with finite second moment.

We thank Reviewer #3 for pointing this out. Since this comment was raised several times in his/her report, we address it in details all in at once.

Tree ansatz (or tree approximation) is one of the most important frameworks to study complex network. The main assumption of this framework is that in the thermodynamics limit there are no finite loops in the network and only infinite loops exist. (Note that the existence of infinite loops is important for the critical behavior. The problem becomes trivial without loops.) Under this assumption we can conveniently use the techniques of random branching processes.

Just as Reviewer #3 mentioned, the approximation becomes exact for networks with a finite second moment of degree, such as Erdős-Rényi (ER) networks and Scale-free (SF) networks with degree exponent $\lambda > 3$. We totally agree with Reviewer #3 on this point. We just want to point out that we can still use the tree ansatz as an approximation to study networks with infinite second moment of degree, such as SF networks with $\lambda \leq 3$. It has been demonstrated in literature that the tree ansatz actually offers very accurate results in various problems on networks such as percolation [2-4], spin model [5-6], graph coloring [7], synchronization [8], and so on. Actually, the tree ansatz has been widely used in studying loopy networks [1]. In this work, we find that it also works very well for studying AP-related problems in SF networks with $\lambda \leq 3$.

A detailed discussion on the tree ansatz is added in the revised SI (page 32, Supplementary Note 4A). We thank Reviewer #3 again for raising this point, which helps us make our presentation more rigorous.

- [1] S. N. Dorogovtsev, A. V. Goltsev, and J. F. F. Mendes, Critical phenomena in complex networks. *Reviews of Modern Physics* 80, 1275 (2008).
- [2] R. Cohen, K. Erez, D. ben-Avraham, and S. Havlin, Resilience of the Internet to random breakdowns. *Phys. Rev. Lett.* 85, 4626 (2000).
- [3] D. S. Callaway, M. E. J. Newman, S. H. Strogatz, and D. J. Watts, Exact solution of percolation on random graphs with arbitrary degree distributions. *Phys. Rev. Lett.* 85, 5468 (2000).
- [4] R. Cohen, D. ben-Avraham, and S. Havlin, Percolation critical exponents in scale-free networks. *Phys. Rev. E* 66, 036113 (2002).
- [5] S. N. Dorogovtsev, A. V. Goltsev, and J. F. F. Mendes, Ising model on networks with an arbitrary distribution of connections. *Phys. Rev. E* 66, 016104 (2002).
- [6] S. N. Dorogovtsev, A. V. Goltsev, and J. F. F. Mendes, Potts model on complex networks. *Eur. Phys. J. B* 38, 177 (2004).

[7] L. Zdeborová and F. Krzakała, Phase transitions in the coloring of random graphs. Phys. Rev. E 76, 031131 (2007).

[8] B. C. Coutinho, A. V. Goltsev, S. N. Dorogovtsev, and J. F. F. Mendes, Kuramoto model with frequency-degree correlations on complex networks. Physical Review E 87, 032106 (2013).

Results:

A more dense section where all definitions and abbreviations are introduced next to each other could serve as reference to return to in the course of reading. The amount of abbreviations is quite substantial and it is easy to lose track.

We thank Reviewer #3 for this excellent suggestion. In the revised manuscript, we added a “Methods” section in the end of the main text (see page 15), where we devoted a particular subsection to introduce all the definitions and abbreviations used in our paper.

Conclusions:

The complex network history and general philosophy could be substantially shortened. Instead, a more detailed summary of the findings and results would be appropriate and would orient a confused reader. What are the actual contributions? What is the line of thought again? Why are APTA and GAPR studied? Which phase transitions occur? Why are they relevant and why do the findings explain the empirical observations obtained with APTA?

We thank Reviewer #3 for this very constructive comment. We have revised the Conclusion section as follows:

In this article, we systematically investigate AP-related issues in complex networks. Many interesting phenomena of APs are discovered and explained for the first time. On the empirical side, we proposed two AP-based applications: a network attack strategy (i.e. APTA) and a network decomposition method (GAPR). We found that, given a limited “budget” (i.e. the number of nodes to be removed), our APTA strategy is more efficient in reducing the GCC of the network than other existing strategies. In revealing the core-periphery structure of complex networks, our GAPR method is quite different from traditional network decomposition methods in the sense that our identified core may include low-degree nodes. Those sparsely connected nodes can be functionally very important, but they are always ignored in traditional decomposition methods. On the theoretical side, we proposed an analytical framework to calculate various AP-related properties, among which the emergence of the RGB as a discontinuous percolation transition is of great theoretical interest. This finding also provides a theoretical explanation of the empirical findings that most of the real-world networks have either a very small RGB or a rather big one.

Taken together, our results offer a new perspective on the organizational principles of complex networks, shed light on the design of more resilient infrastructure networks and more effective destructions of malicious networks, and open new avenues to deepening our understanding of complex networked systems. Since the identification of APs also helps us better solve other challenging problems, e.g. the calculation of determinants of large matrices [44], and the minimum vertex cover problem on large graphs (a classical NP-hard problem) [45], we anticipate that our results on APs will trigger more research activities on those problems as well.

DETAILED POINTS OF CRITIQUE

The abbreviation p. means page, while SI refers to Supplementary Information.

We thank Reviewer #3 very much for his/her careful reading of our manuscript.

Main paper:

-p.1: The analytic framework that you develop does not apply to an arbitrary complex network. You can calculate the average mentioned quantities for configuration model type random graphs where the prescribed degree distribution has finite second moment.

Please see above for our explanation of using tree approximation to networks with infinite second moment of degree distribution. Also, all the calculations in our framework can be applied to networks with infinite second moment of degree distribution.

-p.1: What is a "rich" phase diagram?! You could simply delete this part.

Here "rich" just means that there are many phases in the phase diagram, and hence the phase diagram is "structurally rich". This term has been frequently used in physics literature [1-3].

[1] J. R. Edison, N. Tasios, S. Belli, R. Evans, R. van Roij, and M. Dijkstra, Critical Casimir Forces and Colloidal Phase Transitions in a Near-Critical Solvent: A Simple Model Reveals a Rich Phase Diagram. *Phys. Rev. Lett.* 114, 038301 (2015).

[2] P. Braun-Munzinger and J. Wambach, Phase diagram of strongly interacting matter. *Rev. Mod. Phys.* 81, 1031 (2009).

[3] D. Cellai, E. López, J. Zhou, J. P. Gleeson, and G. Bianconi, Percolation in multiplex networks with overlap. *Phys. Rev. E* 88, 052811 (2013)

-p.2: Please comment that single isolated nodes do not count as component to you (or otherwise if I have not understood correctly).

An isolated node does count as a connected component. Consequently, the only neighbor of a leaf node must be an AP, because its removal will disconnect the leaf node from the network, and cause the leaf to become an isolated node.

-p.2: High potential for shortening. For instance, the sentence "Those AP nodes play an important role in ensuring the connectivity..." is not needed. Its implied by their definition.

In the revised manuscript, we have revised this sentence to "Those AP nodes play important roles in ensuring the robustness and connectivity of many real-world networks". It serves as a statement connecting the definition of AP and the real-world examples.

-p.2: Not nice formulation: "notorious" NP-complete problem

We have replaced "notorious" by "classical" in the revised manuscript.

-p.3: It would be helpful to mention with what kind of attacks to compare APTA with in the SI.

We have mentioned those attack strategies in the main text (see page 2, paragraph 2).

-p.3: "Structural integrity" sounds strange to me.

This term has been used in several papers on complex networks [1-3].

[1] L. Dall'Asta, A. Barrat, M. Barthélemy, and A. Vespignani, Vulnerability of weighted networks. *Journal of Statistical Mechanics: Theory and Experiment* 4, P04006 (2006).

[2] M. Rosas-Casals, S. Valverde, and R. V. Solé, Topological vulnerability of the European power grid under errors and attacks. *International Journal of Bifurcation and Chaos* 17(07), 2465-2475 (2007).

[3] Y.-Y. Liu, J.-J. Slotine, and A.-L. Barabási, Controllability of complex networks. *Nature* 473, 167-173 (2011).

-p.3: Please, clarify in the paper: How does GAPR precisely work? If two APs are connected so that the removal of one would be enough to split the network in two components, do you still remove both APs? I guess so from the text.

Correct. In the revised manuscript, we have added the following sentence to clarify this point (see main text, page 3, 2nd paragraph):

Note that at each step, we simultaneously remove all the APs present in the current network.

-p.4: "still lack a deep understanding on the roles..." -> "still lack a deep understanding of the roles..."

Fixed.

-p.4 (Analysis of Real Networks): "relative size": Please, add relative to the total number of nodes in the network

In the revised manuscript (see main text, page 5, 1st paragraph), we gave the exact definitions of $n_{AP} (:=N_{AP}/N)$ and $n_{RGB} (:=N_{RGB}/N)$, and added the following sentence:

Here N_{AP} , N_{RGB} , and N represent the number of APs, the number of nodes in the RGB, and the number of nodes in the whole network, respectively.

-p.4 (Analysis of Real Networks): "relative size": I personally find the notation $n_{\{*\}}$ a bit unfortunate, because I would think you refer to a number. What about r or ρ or d ?

Here we are following the notation convention in physics literature, where capital letters, e.g., N , stand for numbers, where small letters, e.g., n , represent density, fraction, etc.

-p.5: "Ironically" sounds strange to me in a paper.

We have replaced it by "Interestingly".

-p.5: Comment: It's not surprising to find the mentioned result. The studied (infrastructure) networks are not optimized with respect to APTA, of course. They basically lack a high redundancy, because building a link comes with a high cost. (Still, the networks are often optimized with respect to other criteria that make sense.) Of course, you can suggest to take your considerations into account to build more robust systems. However, you would need to argue why you only care about the GCC and not the size of the other components. Maybe, some of your networks are able to maintain a certain functionality without being connected to the largest component. Or, also the largest component cannot function without the rest, because a critical node was removed. The functionality of each node and how they contribute to the whole functionality would decide about the answers to these questions. Still, you present a nice proxy and the motivation to study GAPR processes is clear.

We thank Reviewer #3 for this very insightful comment. In the revised manuscript (see main text, page 5, 2nd paragraph), we have added the following sentences:

These results suggest that infrastructure networks are apparently not optimized with respect to AP removal. Indeed, due to the high cost of adding new links (e.g. connecting two power stations with high-voltage transition lines, or connecting two cities with a new highway), infrastructure networks typically lack a high redundancy, but are often optimized with respect to other criteria, such as social profitability.

.....

In other words, those ecological networks tend to be biconnected and the extinction of one species will not disconnect the whole community. This high structural robustness could be due to evolutionary inter-species interactions across the whole community [19].

[19]: How Do Species Interactions Affect Evolutionary Dynamics Across Whole Communities?
Timothy G. Barraclough, *Annu. Rev. Ecol. Evol. Syst.* 2015. 46:25–48

-p.5: In what respect is the phase transition "novel"?

We call these phase transitions novel simply because they have never been reported before, and the underlying process (i.e. GAPR) is also novel.

-p.5: Not all deterministic processes can be described analytically. What is really important to make GAPR tractable is that you get rid of the optimization of damage that an AP causes in APTA.

We fully agree with Reviewer #3. We have added this point to the revised main text (see page 5, 1st paragraph).

-p.6: The model with prescribed degree sequence/degree distribution is called configuration model (as you name it later) and goes back to:

```
@article{Molloy1995,  
author = {Molloy, Michael and Reed, Bruce},  
journal = {Random structures {\&} algorithms},  
number = {1995},  
pages = {161--179},  
title = {{A critical point for random graphs with a given degree sequence}},  
volume = {6},  
year = {1995}  
}
```

We fully agree with Reviewer #3. Actually, this important paper has been cited in the previous version of our paper as Ref [19]. In the revised manuscript, the paper was cited in an even earlier place (page 7, 1st paragraph, 1st sentence).

-p.6: Some of your networks are small enough that the simulation results lead to rather broad distributions. This means that not all networks with the given prescribed degree distribution have the same properties and some in the sample also have degree-degree correlations, for instance. Thus, you cannot simply attribute the properties that you observe to the degree distribution in all of these cases.

We fully agree with Reviewer #3. Just as we mentioned in the last part of the subsection “Analysis of Real Networks”, higher-order structure correlations, such as clustering and degree assortativity also contribute to the number of APs and the size of RGB. However, the degree distribution is the most important factor that determines those properties.

Usually, we cannot consider every aspect in the systems at the same time. In these cases, a simplified and tractable model that captures the most important feature is already helpful to understand the system. In our work, this feature is degree distribution. However, our framework can also be used to study other features, such as degree-degree correlation. But we consider it is beyond the scope of the current work. And we leave it as a future work or a community effort.

-p.6 (Results): What about a transitional sentence that now you move the attention now to

infinitely large networks and a more in depth exploration with respect to different degree sequences?

We thank Reviewer #3 for this excellent suggestion. We have added a transitional sentence at the beginning of this paragraph (see main text, page 7, 2nd paragraph)

The results of real-world networks prompt us to analytically calculate n_{AP} and n_{RGB} for networks with prescribed degree distributions²³. In the subsection, we analyze the GAPR process on infinitely large networks and explore in depth the effect of different degree distributions on n_{AP} and n_{RGB} .

-p.7: State your assumptions more clearly: Infinitely large networks, finite second moment of degree distribution required, in which order do you take the thermodynamic limit and $c \rightarrow \infty$?

Please see above for our response to the concern about networks with infinite second moment of degree. Regarding the order of taking limits, since all the discussion in theoretical part is about infinite large networks, we take the thermodynamic limit first.

-p.7: $\lambda > 3$ or you need a finite cut-off degree.

Please see above for our response to the concern about networks with infinite second moment of degree.

-p.7: c is not generally defined as average degree. It is just introduced as parameter of the Poisson degree distribution.

We thank Reviewer #3 for pointing this out. We have added a statement to clarify this point (see page 8, 2nd paragraph)

... hereafter, we also use c to denote the mean degree of a general network...

-p.7: In general, you do not need to give so many digits of c_{AP} (or other numerical values). Two are fully enough.

We agree with Reviewer #3. Here we use more digits simply to emphasize that those numbers are typically not rational. Again, we are following conventions in physics literature [1-3].

[1] D. Stauffer, and A. Aharony, Introduction to percolation theory. CRC press, 1994.

[2] S. N. Dorogovtsev, A. V. Goltsev, and J. F. Mendes, Critical phenomena in complex networks. Reviews of Modern Physics 80, 1275 (2008).

[3] M. E. J. Newman, The structure and function of complex networks. SIAM review 45, 167-256 (2003).

-p.7: Please, cite a supporting article for $c_p = 1$ for percolation in ER networks.

Done. Thanks for the reminder.

-p.8: Does your argument for ER networks with adding links also apply to SF networks? Isn't it important which nodes with which degree receive higher degrees? Refer also to SI for explanation how the increase of average degree works.

We thank Reviewer #3 for those excellent questions. Yes, we can still use our argument of adding two types of links to explain the non-monotonic behavior of $n_{AP}(c)$ in SF networks. Yet, as Reviewer #3 kindly pointed out, the degree heterogeneity should come into play. Indeed, in the static model of SF networks, the nodes are labeled from 1 to N , and node i is assigned a weight $p_i \sim i^{1/(\lambda-1)}$, ($i = 1, \dots, N$). Then the process of adding links is as follows: We randomly choose two nodes based on their assigned weights; we add a link between these two nodes if they have not been connected; we repeat this process until a preset number of links have been added to the network (so that the network has a desired mean degree c). Note that those nodes with larger weights tend to have higher degrees in the generated network. In other words, those large-weight nodes tend to be hubs.

In the revised manuscript, we have added the following discussion into the Supplementary Information (see Supplementary Note 5 - Behavior of c_{AP} for SF Networks).

According to the process of adding links in the static model, we can qualitatively explain the behavior of $n_{AP}(c)$ and c_{AP} for SF networks with degree exponent λ generated by the static model. Compared with ER networks where links are added uniformly, in SF networks generated by the static model the link density between hubs is much higher. This important feature directly leads to three properties: (1) As we add more links, those hubs are quickly connected to each other and form a giant component, which results in smaller percolation mean degree $c_p(\lambda)$ than that of ER networks. (2) Since a large fraction of nodes in the static model have small weights (and degrees), the link density between them is low. For $c > c_p(\lambda)$, except the GCC, the rest of the network is so fragmented that the sizes of the FCCs are very small. Note that all the FCCs are trees in thermodynamics limit, because the probability of adding a link between two nodes with small weights in the same FCC to form a loop is nearly zero. (3) Since the FCCs are very small, in the process of adding links the GCC grows very slowly. It indicates that compared with ER networks, we have to add more links for the GCC to reach a critical size so that the contributions of type-I links (links inside the GCC) overwhelms that of type-II links (links that connect the GCC with an FCC or connect two FCCs), which results in a larger $c_{AP}(\lambda)$. The above properties explain why the distance between $c_p(\lambda)$ and $c_{AP}(\lambda)$ is larger than that for ER network, as well as the observation that $c_{AP}(\lambda) > c_p(\lambda)$ is more prominent for smaller λ .

-p.8: Add one step in the argument in the second paragraph: For SG networks: GCC's relative size is rather small..., - and the network is more fragmented in small components - which results in a larger c_{AP} . -> Why does it really need to? Doesn't it depend on how the components are connected among each other and not their size or number? For instance, many (small or large)

fully connected cliques do not have any APs. It really needs to have something to do with the high degree heterogeneity.

We thank Reviewer #3 for this excellent suggestion. To better explain the behavior of $c_{AP}(\lambda)$ for SF networks, we have revised that paragraph as follows (see the revised main text, page 9, 2nd paragraph):

The phenomena that $c_{AP} > c_P$ is even more prominent for SF networks, where $c_P(\lambda) < 1$ and $c_{AP}(\lambda) > 1.41868 \dots$. This is because, for those SF networks, even though the GCC emerges at lower $c_P(\lambda)$, its relative size is rather small at the initial stage of its emergence and the network is more fragmented in FCCs, which results in larger $c_{AP}(\lambda)$ (Fig. 3b) (See Supplementary Note 5 for details).

As we mentioned in the previous response, in the thermodynamic limit, those small FCCs cannot be cliques, because the probability of adding a link between two nodes with small weights in the same FCC to form a loop is nearly zero.

-p.9: In general not a too nice formulation in a paper: "which is nothing but" -> i.e.

Fixed.

-p.9 or 10: Another topic sentence as introduction would be nice. A small summary of what you are going to present and why you analyze the specific variables that you analyze.

Done (see main text, Page 10, 1st paragraph). The added topic sentence reads as follows.

In order to systematically characterize the percolation transitions, various quantities will be analyzed, such as the critical mean degree, critical exponents, the jump size of the order parameter at criticality, and so on.

-p.13: The results are not discussed at all! This would be a good opportunity to forge a bridge to the introduction and discuss the implications of your work.

Following this excellent suggestion, we have completely rewritten the Conclusion section (see revised main text, page 14).

-Graphics:

Considering the space that you use for all the plots, you could make them much larger and better readable. Some color drawings almost vanish when you print the paper. Define: abbreviation s.e.m.

All the figures have been rescaled in the revised main text to make them larger and better

readable. We also adjusted the color of the figures in Illustrator for clearer appearance in printed form.

The definition of s.e.m. has been added in the figure caption.

Supplementary information:

-p.4: The name configuration model could be mentioned again.

Done.

-p.4/6: Please, clarify the GAPR: Do you remove APs that are connected to each other? (I guess that yes.)

Yes, in each time step, we remove all the APs, no matter how they locate in the network. In the revised SI, we have added the following sentence to clarify this point (see revised SI, page 29):

Step-2: Simultaneously remove all the APs and the links attaching to them in the current network.

-p.7: Not everyone knows what a thermodynamic limit is. Please explain that you study the limit $N \rightarrow \infty$.

Done. Thanks for your reminder.

-p.7: The method that you use is also called Local Tree Approximation/ Branching Process Approximation/ Heterogeneous Mean Field Approximation. It's not really a tree ansatz, because you only consider locally tree-like structures, not real trees. This makes a huge difference. In a tree, every node with degree > 1 would be an AP.

Please see our above response. We use local tree approximation instead of tree ansatz. We want to mention that in physical literature tree ansatz is the same as tree approximation.

-p.7: Important: The locally tree like structure only appears for finite second moment of the degree distribution! This also explains why your numerical calculations do not match perfectly the simulations for $\lambda \leq 3$.

Please see our above response.

p.7: Last paragraph in "Tree ansatz": See also:
@article{Melnik2011,

```

author = {Melnik, Sergey and Hackett, Adam and Porter, Mason a. and Mucha, Peter J and
Gleeson, James P},
doi = {10.1103/PhysRevE.83.036112},
issn = {1539-3755},
journal = {Physical Review E},
number = {3},
pages = {36112},
title = {{The unreasonable effectiveness of tree-based theory for networks with clustering}},
volume = {83},
year = {2011}
}

```

We thank Reviewer #3 for pointing out this important paper. We have cited it in the revised manuscript.

p.7: Definition of type of nodes together with Eq. (2) on p.8: I did not understand what you mean from the text that you write on p.7. On the basis of Eq. (2). I guess, you mean: β_t -nodes: nodes that are identified as AP nodes exactly at time t and belong to the GCC at time t . γ_t -nodes: nodes that belong to the GCC all the times $\{0, \dots, t\}$. State explicitly that α_t also includes APs.

In the revised SI (page 33, 2nd paragraph), those definitions have been presented in more detailed way as follows:

1. α_t -nodes: nodes that belong to FCCs at time t (note that by definition α_t -nodes include APs in FCCs);
2. β_t -nodes: nodes that are APs in the GCC at time t ;
3. γ_t -nodes: nodes that are not APs and belong to the GCC at time t . Note that if a node is a γ_t -node at time t , this node must be γ_τ -node with $\tau < t$.

p.7: β_t and γ_t are not defined as probabilities on p.6, when you define $\beta_t + \gamma_t$!

Actually we use the state of a node and the probability of the node in that state in an exchangeable manner. We have added this following sentence to the revised SI (see page 33):

Note that for convenience sake here we use the same notation to denote both the state of a node and the probability of a node in that state.

p.8: A sentence would help why you focus on probabilities of the end node type of a link first. The goal is to calculate n_{AP} and N_{RGB} etc. There, you need to know the type of neighbors of a node to judge the state of a node. Thus, you calculate the probability first that neighbors of a node (i.e. the end nodes of a randomly drawn link) are of a certain type.

We fully agree with Reviewer #3. The following sentence has been added into revised SI (page 33).

In order words, in order to determine the state of a node, we need to know the states of its neighbors.

-p.9: Fig.8 is pretty trivial and not needed from my point of view. The introduction of notation could be added to the next figure.

We consider that presenting this figure separately is necessary because we not only define the diagrammatic representations of $\alpha_t, \beta_t, \gamma_t$, but also define the diagrammatic representation of the probability that an end node of a randomly chosen link belongs to the GCC and the manipulations of those probabilities.

-p.9: Tree ansatz not good formulation.

In the revised SI, this term has been changed to local tree approximation.

-p.9: It is absolutely crucial for your derivations that the removal of an AP does not break down the GCC in two components of non-zero relative size. This needs more explanation or at least a reference for a proof.

Indeed, this is one of the important properties for a locally tree-like network, i.e., there can be only one GCC in the network. In the revised SI (page 32, Supplementary Note 4A), we explicitly mentioned this point with a reference paper.

-p.11: Technically, $G_1(x)$ is not the generating function of $Q(k)$. This would be $\sum_k Q(k)x^k$. $G_1(x) = G'_0(x)/c = G'_0(x)/G'_0(1)$, where $G_0(x) = \sum_k P(k)x^k$.

We fully agree with Reviewer #3. In the revised SI (page 35), that sentence has been revised as follows:

“...is the generating function of the branching processes [48].”

-p.12: State more clearly: ...: Fraction of APs whose removal splits the GCC in a smaller GCC and FCCs ...: Fraction of APs whose removal splits an FCC in a smaller FCCs.

In the revised SI (see page 36), this point has been mentioned as follows:

1. $n_{AP}^{GCC}(t, c)$: fraction of APs that belong to the GCC whose removal splits the GCC into a smaller GCC and many FCCs;
2. $n_{AP}^{FCC}(t, c)$: fraction of APs that belong to FCCs whose removal splits FCCs into smaller FCCs.

-p.14: State why you write the formulas in terms of generating functions... Eases calculations later, when you have explicit close form solutions for G_0 .

Yes, it is easy for calculation to write the equations in terms of generating functions. The statement has been added (see revised SI, page 38).

-p.17: Definition of a : You mean $a = 1/(\lambda - 1)$, I guess.

Correct. This typo has been fixed.

-p.20 and following: As written again. Consider that your analytic derivations only hold for degree distributions with finite second moment. The cases of SF with $\lambda = 3$ and $\lambda = 2.5$ do not need to work theoretically.

Please see our above response.

Overall, the work is very interesting and a deep analysis is provided. However, considering the high amount of material, it needs additional effort to present it in an more easily accessible way.

Finally, we thank Reviewer #3 again for his/her extremely careful reading, very insightful and constructive comments, which help us significantly improve our presentation. We hope our responses above have addressed those very legitimate issues/concerns in a satisfactory manner.

Reviewer #1 (Remarks to the Author):

The authors have addressed all my comments.

Reviewer #2 (Remarks to the Author):

The authors have properly addressed all raised questions.

Reviewer #3 (Remarks to the Author):

I see a significant improvement of the manuscript and do not object the choice of the authors to keep their overall paper organisation. Especially the conclusion has the relevant content now.

My points of critique have been mainly addressed, except one:

The authors correctly mention now that the local tree approximation (LTA) is only exact for degree distributions with finite second moment (even though numerical approximations are still good in a number of cases when this condition is not satisfied) in the supplementary material. This information needs also be added to the main paper, where the LTA approach is sketched. The information is important for researchers who want to apply this method in a similar manner and informs about the constraints of the approach. This knowledge cannot be taken for granted for readers of an interdisciplinary journal.

Response to Reviewer #3

I see a significant improvement of the manuscript and do not object the choice of the authors to keep their overall paper organisation. Especially the conclusion has the relevant content now.

We thank Reviewer #3 for his/her positive assessment of our revised manuscript.

My points of critique have been mainly addressed, except one:

The authors correctly mention now that the local tree approximation (LTA) is only exact for degree distributions with finite second moment (even though numerical approximations are still good in a number of cases when this condition is not satisfied) in the supplementary material. This information needs also be added to the main paper, where the LTA approach is sketched. The information is important for researchers who want to apply this method in a similar manner and informs about the constraints of the approach. This knowledge cannot be taken for granted for readers of an interdisciplinary journal.

We thank Review #3 for this very constructive suggestion. We have added a paragraph in the Methods section to introduce LTA and its constraints (see main text, page 16, 2nd paragraph):

Our theoretical treatment of the GAPR process is based on the local tree approximation, which assumes in the thermodynamics limit (i.e. network size $N \rightarrow \infty$) there are no finite loops in a network and only infinite loop exist^{23-25,28}. This approximation allows us to use the convenient techniques of random branching processes to solve the GAPR process on large uncorrelated random networks (Supplementary Note 4). Note that the local tree approximation is only exact for networks with finite second moment of the degree distribution. However, it has been demonstrated in various network problems that this approximation can obtain very accurate results even for networks with diverging second moment of the degree distribution²⁸. Here we find that this local tree approximation works very well in analysis of the GAPR process (see Supplementary Notes 4 and 6).